# Antimicrobial Resistance in Veterinary Medicine: An Overview

**DOI:** 10.3390/ijms21061914

**Published:** 2020-03-11

**Authors:** Ernesto Palma, Bruno Tilocca, Paola Roncada

**Affiliations:** Department of Health Science, University “Magna Graecia” of Catanzaro, Viale Europa, 88100 Catanzaro, Italy; palma@unicz.it (E.P.); tilocca@unicz.it (B.T.)

**Keywords:** antibiotics, antimicrobial resistance, domestic animals, microorganism

## Abstract

Antimicrobial resistance (AMR) represents one of the most important human- and animal health-threatening issues worldwide. Bacterial capability to face antimicrobial compounds is an ancient feature, enabling bacterial survival over time and the dynamic surrounding. Moreover, bacteria make use of their evolutionary machinery to adapt to the selective pressure exerted by antibiotic treatments, resulting in reduced efficacy of the therapeutic intervention against human and animal infections. The mechanisms responsible for both innate and acquired AMR are thoroughly investigated. Commonly, AMR traits are included in mobilizable genetic elements enabling the homogeneous diffusion of the AMR traits pool between the ecosystems of diverse sectors, such as human medicine, veterinary medicine, and the environment. Thus, a coordinated multisectoral approach, such as One-Health, provides a detailed comprehensive picture of the AMR onset and diffusion. Following a general revision of the molecular mechanisms responsible for both innate and acquired AMR, the present manuscript focuses on reviewing the contribution of veterinary medicine to the overall issue of AMR. The main sources of AMR amenable to veterinary medicine are described, driving the attention towards the indissoluble cross-talk existing between the diverse ecosystems and sectors and their cumulative cooperation to this warning phenomenon.

## 1. Antibiotic Resistance: Origin and Diffusion

Nowadays, antimicrobial resistance (AMR) is intended as a “modern” microbial feature resulting from the unsuccessful and/or prolonged exposure to antibiotic treatments. However, the identification of antimicrobial resistance traits in ancient permafrost, isolated caves, and mummies witness the presence of the antimicrobial features since remote times, and a distinction between innate and acquired AMR is currently employed. The first is the result of a slow and long evolutionary process that microorganisms have performed to adapt to the changing environmental conditions; while the second is the result of a “quick” adaptation to a sudden selective pressure represented by the antimicrobial treatment [1,2].

Intrinsic antimicrobial resistance is a peculiarity of both pathogenic and non-pathogenic microorganisms since required for the survival and evolution of the bacteria in a dynamic environment [3,4,5]. Major mechanisms of the intrinsic bacterial resistance rely on the impermeability of the bacterial cell to the antibiotic molecule (e.g., physicochemical properties of the molecules, presence efflux pumps etc.), the lack of the target molecules or the inactivation of the antibiotic compound by means of degrading enzymes [5,6]. Altogether, this is the result of the concerted activity of both genetic/heritable elements and other phenotypic traits involved in a wide array of metabolic functions, which only recently have been considered as pivotal for the onset and diffusion of the AMR, Figure 1.

Following environmental changes, bacterial evolutionary machinery makes use of their genomic flexibility to better suit the surrounding environment including, among others, the ability to protect themselves from toxic substances [7]. Moreover, the genetically determined resistance (i.e., the ensemble of genes involved in the AMR) set up by given bacteria is efficiently transmitted to its clonal expansion and/or other bacterial specimens through mobile genetic elements, such as plasmids, transposons, and integrons [4].

Plasmid-induced resistance accounts for the greatest portion of resistance spread. Despite the common thought of a rapid and easy loss of the resistance-encoding elements when no selective pressure is exerted by the antibiotic treatments, the recent assessment of genetic stability of plasmids along with the recurrent identification of resistance plasmids in natural isolates of microorganisms (mainly bacteria) suggest plasmids as a very important reservoir of AMR genes that can be propagated over the time and across species borders effortlessly and in a time-effective manner [4,8,9,10].

Transposons are responsible for the microbial genome flexibility to a wide extent. Transposable elements are capable of changing their position within the same DNA molecule or jump between diverse DNA molecules, including plasmids. This, in turn, results in the alteration of the whole genetic background to an extent depending on thelocus where the transposable elements are inserted [11,12,13,14]. Although transposition might lead to deleterious mutations/effects, the genetic composition of transposons and insertion sequences may include AMR genes; hence, confer and propagate survival traits under harsh conditions, acknowledged its proper insertion in the bacterial genome.

Integrons are associated with great benefits for bacterial fitness and robustness, providing valuable “tools” that enable survival under extremely varying environments. Among the array of genetic elements contained in integrons, the AMR genes have shown to be of great importance, supporting bacterial survival under antibiotic treatment and/or the co-existence of diverse bacterial members of a heterogenous consortium. Analogously to the other mobile genetic elements, integrons can be mobilized both by the bacterial chromosome and plasmids and can be propagated and integrated far from their original site, conferring antimicrobial protection to a wide number of microorganisms [4,15].

Another less-known mobile element system employed by bacteria for horizontal gene transfer is represented by the Integrative and Conjugative Elements (ICEs). The system has been relatively recently described as one or more self-transmissible integrative elements [16], capable of own mobilization to carry several bacterial traits such as virulence factors, biofilm formation capability [17,18,19], resistance to heavy metals [19], and antibiotic resistance genes [4,20]. Unlike plasmids, ICEs are not extrachromosomal elements since lacking an autonomous replication origin; however, the wide array of traits potentially propagated by these elements are attractive for an increasing number of research groups and a comprehensive integrated database on the ICEs has been released in 2012, ICEberg [21].

Bacterial toxin-antitoxin systems (TAS) have also been related to antibiotic resistance. TAS is mostly identified in the bacterial specimens subjected to antibiotic treatment [22,23]. When carried in a plasmid, the bacterial toxin-antitoxin system represents a vertical way for the transmission of the survival traits, but it can also be present in the chromosomal DNA and it is thought to perform several complex cell functions, including AMR; however, the mechanisms exerted by TAS are not fully elucidated and opposing views are reported in the literature concerning the mode of action of TAS while conferring antimicrobial persistence in bacteria. Studies demonstrated that activation of the TAS results in the production of persisters, i.e., dormant cells capable of surviving the harsh conditions activating TAS. In this view, Jurenas and colleagues reported the stimulation of persisters cells following the alteration of transfer RNA (tRNA) functions [23]. TAS activation in *Salmonella enterica* generated drug-tolerant strains [24]; whereas, chromosomal TAS has been activated by environmental stressors [25]. On the other hand, several studies opposed to the hypothesis of TAS involvement in persister cells production, suggesting TAS having a pivotal role in other processes of the bacterial biology [26,27,28,29].

The study and mapping of the genetic elements carrying AMR traits is of paramount importance for monitoring and predicting resistance evolution over the time and space; however, these are not the unique players of the AMR since other non-mobilizable non-heritable elements such as growing in biofilm, swarming adaptation and persister cells are important contributors to the antimicrobial-resistant phenotype [5,6,30].

Phenotypic persistence is the condition where bacterial cells are not killed by the dose/type of antibiotic administered; instead, cells are in a quiescent status that is reversed once the stressor is removed. Nevertheless, the antibiotic susceptibility is reverted along with the bacterial growth and second exposure to the antibiotic results in the bactericidal or bacteriostatic effect [31]. This is because the antibiotic resistance is not determined by genetic mutation and the genotype of the persisting cell is the same as the “wild type” [5]. Anyway, the presence of persisters cells during infections results in a failure or diminished efficacy of the antibiotic-based treatments, posing the need for further investigations on the molecular mechanisms driving the insurgence of persistent phenotypes and the design of therapeutic interventions based on the eradication of antibiotic-induced persistent cells. Besides TAS, other bacterial features modulate the development of persistence. These include elements that alter the metabolism regulation and affect the bacterial gene expression profile such as flagella encoding genes, energy production enzymes, and other genes related to cell growth rate [32,33].

Biofilm growth has shown to preserve bacterial cells from antibiotic treatment [34]. This has a clinically relevant meaning in all the chronic infections and the potential colonization of surfaces (e.g., intubated patients, prothesis, catheters). Biofilm formation is rather easy and might represent a treat/complication for the patient’s infection. The mechanisms responsible for the antibiotic resistance in the biofilm-growing colonies rely on the properties of the biofilm matrix and the metabolic behavior of the microbial community. In the first case, the matrix might reduce antibiotic efficacy by diminishing its diffusion within the bacterial biomass or by seizure of the antibiotic molecules through its complex structure. In the second case, it is worthy of note that different regions of the biofilm biomass are featured by subpopulations of bacteria with a peculiar metabolic feature, mostly due to the changing physicochemical condition (e.g., gradient of nutrient availability and gradient of oxygen). This implies the presence of bacterial populations with a dynamic susceptibility to the diverse types of antibiotics. As an example, it has been demonstrated that oxygen-rich regions of *Pseudomonas aeruginosa* biofilms are susceptible to quinolones while cells in the same regions are resistant to cationic peptides. The opposite trend was observed in the low-oxygen regions of the biofilms [35], indicating that the phenotypic resistance of bacterial biofilms depends on several factors that operate in a complementary manner.

Another interesting strategy employed by bacteria to shield themselves from antibiotic treatments is the swarming movement. Although it might appear as a “sole” type of movement of a bacterial population, this encompasses complex physiological adaptation processes expecting several genes changes, including the increased codification for porins and efflux pumps. These, in turn, might lead to reduced antibiotic activity on the target bacterial cells [36]. Furthermore, several proteases affecting swarming motility are relevant for the formation of biofilms, and therefore involved phenotypic antibiotic resistance [6,37,38]. Nevertheless, the molecular mechanisms leading to the antibiotic resistance are still far to be fully understood, since other elements might be involved in the antimicrobial-resistant phenotype. Intensive research programs are being performed worldwide to gain further knowledge on the variety of mechanisms taking part in AMR. Quorum sensing, for instance, other than being involved in the biofilm production, is also responsible for the cellular reprogramming that occurs in swarmer cells [39,40]. Moreover, the recent uncovering of moonlighting proteins, i.e., proteins capable of different and independent functions as a function of their localization, opens new avenues in the investigation of the AMR mechanisms. Here, the pivotal contribute to bacterial virulence might also have a role in the AMR, especially when bacterial cells are considered as members of wider and heterogeneous microbial community [41,42,43]; thus, enabling a clearer depiction of the refined tuning existing between microbiota composition and its functional shaping in response to exogenous stressors like the antibiotic-based treatments.

## 2. One-Health Approach to Face AMR

Being ubiquitarians, microorganisms represent a pool of AMR traits in all ecological niches. The complex network of interactions occurring between microbial specimens from diverse “environments” facilitate the gene flow, expanding the AMR between humans, animals, and the environment, resulting in an overall issue. Thus, a coordinated multisectoral approach such as One-Health is desired to investigate and address this warning phenomenon [44,45]. It is defined as One-Health, “the collaborative effort of multiple health science professions, together with their related disciplines and institutions -working locally, nationally, and globally -to attain optimal health for people, domestic animals, wildlife, plants, and environment” [46]. In such a comprehensive approach, the antimicrobial use (and misuse) in the human, animal, and environmental sectors, along with the global-scale spread of the resistance mechanisms within and between these sectors are identified as the major AMR driving forces [47]. Most of the classes employed in the treatment of human infections are shared with the veterinary sector, resulting in a cumulative selective pressure exerted to the microorganisms; thus, reduced efficacy of the antimicrobial-based treatments in both human, veterinary and environmental field [48]. In the human sector, the main actions undertaken by the One-Health approach include a higher consciousness while prescribing antibiotic treatment, preventing over-prescription and improvement of the hygiene conditions and infection control plans. The One-Health actions in regard to the environment sector include the appropriate treatment of industrial, civil and farm waste, on the attempt to reduce the overall dissemination of the AMR traits between sectors [46]. Acknowledged the multifaceted and wide extension of the One-Health approach, this manuscript focuses on the measures actuated by the One-Health approach in the animal sector. These consist of evaluating the impact of the companion animal population and the human-animal relationship; the aquaculture influence on the environment and the human/animal treatment efficacy and the reduction of mass medication in the animal herds other than the need for a global policy that regulates the use of antibiotics as growth-promoting factors in the diverse animal producing countries.

## 3. The Contribution of Companion Animals in AMR Spread

In industrialized countries, the companion animal population is dramatically increased during the last decades and its trend expects steadily increasing growth [49]. The role of companion animals is changing accordingly. The increased attention level towards the animal health status and hygienic conditions results in the overall improvement of the animal welfare [49]. On the other hand, the increased “contact” between animals and human beings lead to a higher risk of infections and the cross-transmission of AMR traits. Thus, the potential of reverse zoonosis along with the creation of animal reservoirs that keep the loop of infection and AMR diffusion is, nowadays, gaining a steadily increasing concern [50,51].

Antimicrobial resistance of pet origin, responsible for both direct and/or indirect threat on the human health, regard principally Methicillin-Resistant Staphylococcus aureus (MRSA), methicillin-resistant staphylococci, vancomycin-resistant enterococci, carbapenemase-producing enterobacteria and Extended Spectrum Beta-Lactamase (ESBL) Gram-negative bacteria [52]. To mention an example, back in 2003–2004, a 24-month screening survey was performed to determine the frequency of methicillin-resistant *Staphylococcus spp* in companion animals. The study, based on 20,366 dogs and 8026 cats, highlighted the alarming prevalence of methicillin resistance traits with 40% of the resistant Staphylococcal population identified as *Staphylococcus schleiferi*, 35% *S. aureus*, and 17% *Staphylococcus intermedius*. The isolation of *S. schleiferi* was more recurrent in dermatitis and ear canal infection, mostly on dogs. Similarly, resistant *S. intermedius* strain was prevalently isolated from dogs; whereas, MRSA isolation was associated with deep infections with a similar frequency in dogs and cats. Tests for resistance to other classes of antibiotics indicated MRSA as resistant to the most commonly employed antimicrobials, followed by resistant *S. intermedius* strain, while *S. schleiferi* resulted as the most susceptible one [53]. These observations were mainly attributed to the increased antimicrobial treatments of companion animals, indicating antibiotics over-usage as the leading cause of the AMR onset and diffusion. Nowadays, several epidemiological studies are being performed to monitor the prevalence and dissemination of AMR. Nevertheless, these epidemiological parameters cannot be clearly defined since influenced by several variables including the population features, geographical location, and investigative methods employed for the survey [52,54]. Regardless of the epidemiological data, accumulating evidence suggests that investigation of the routes undertaken by microorganisms to manifest and/or transmit the AMR are worth of effort. In this view, the sole abuse/misuse of antibiotics is not enough for such a massive transmission of resistant microorganisms between human and pets [55]. Several research lines are being explored, such as the human-animal transmission and vice versa, although controversial results are being observed [56,57,58,59]. Moreover, a pivotal contribution in the AMR dissemination is most likely provided by the environment, intended as the vector connecting the human and animal environment, including the anthropic activities [55]. Moreover, it is believed that monitoring of non-pathogenic specimens and their potential capability to acquire resistance traits is a promising strategy to predict (and prevent) the future resistant strains.

## 4. Aquaculture and AMR

Aquaculture production accounts for almost half of the fish and fish-by product consumed worldwide, posing the need to move towards intensive and semi-intensive production practices [60]. Consequently, antibiotics usage for both therapeutic and nontherapeutic purposes is dramatically increasing. Sulfonamides, penicillins, quinolones, tetracyclines, and phenicols are the antibiotic classes of most concern and their extensive usage is yet associated with a significant contribution to the overall spread of AMR between all animal species, humans included [60,61].

Analogously to the terrestrial intensive production system, the rearing of animals in small spaces increases the overall stress status and the incidence of infective diseases, resulting in whole stock losses associated to economically important repercussion [62]. This unavoidably leads to the customary employment of antimicrobials for both prophylactic and therapeutic purposes, resulting in a strong selective pressure that favors the emergence and selection of AMR strains, and the subsequent dissemination of the AMR traits through mobilizable elements via various routes (food, feed, environment) [60,63,64,65]. To date, no antibiotics have been specifically designed for the exclusive use in aquaculture; thus, antimicrobial compounds destined to other sectors of the human and veterinary medicine are improperly used in the aquaculture context, enhancing dramatically the impact of AMR onset and diffusion [60,61]. Six of the antibiotic classes listed by the World Health Organization (WHO) as critically important for human medicine (aminoglycosides, macrolides, penicillins, quinolones, sulphonamides, and tetracyclines) are commonly employed in both terrestrial and aquaculture husbandries, resulting in an enormous contribution to the reduced efficacy of such compounds in the treatment of human-relevant infections [60,61,66,67]. Furthermore, inadequate usage of antibiotics is associated with a reduced capability of the fish species to effectively metabolize the administered drugs. Thus, antibiotic residues persist for a prolonged time in the fish meat, facilitating the diffusion to the terrestrial ecosystem via the food chain. In addition, it has been estimated that 70–80% of the active compounds are eliminated with faeces enabling antibiotics dispersion through wastewater; thus, influencing a plurality of ecosystems [66,68].

## 5. Domestic Animal Husbandry Influence on AMR

Since their breakthrough discovery, antibiotics held the promise of treating and controlling infectious diseases, triggering the massive rise of antibiotic usage in all applicative fields, including the common animal husbandry practices [69,70].

Traditionally, animal husbandry made use of antibiotics for the treatment of infectious diseases, but also in the design of prophylactic measures and as growth promotor factors [71,72]. The latter application relies on the previous observations linking the administration of subtherapeutic doses of antibiotics to a significant weight gain among the treated animals [73]. Although a clear mechanism for this phenomenon has not yet been fully understood, it has been observed that a prolonged administration of antibiotics at subtherapeutic doses target multiple organs and physiological processes resulting in (i) a reduced biodiversity of the intestinal microbiota and a reduced competition for the nutrients, (ii) reduced number of harmful bacteria, (iii) reduced immune stimulation, (iv) increased biosynthesis of vitamins at intestinal level, and (v) altered metabolism [74,75]. Altogether, these results in an improved net energy balance, thus better animal performance under the zootechnical point of view. Nevertheless, antibiotic administration as growth-promoting factor is unable of irreversible destruction of the harmful bacteria [72]. Moreover, sublethal doses of antibiotics work as selective pressure stimulating the bacterial evolutionary machinery to adapt to the environmental stressors and allowing the more resistant specimens to survive and propagate the AMR traits. To prevent this issue, the European Union enacted the absolute ban of antibiotics usage as growth-promoting factors, since 2006.

Analogous effects are amenable to the use of antibiotics for prophylactic purposes [76,77]. In this view, antimicrobial compounds were commonly administered with the drinking water or the feed ration, ensuring a prolonged exposition of the animals to the low dosage of antibiotics for a prolonged time. However, the protective effects are reverted with the suspension of the antibiotic administration and animals remain susceptible to the infections [72].

Prophylactic-oriented administration of antibiotics in cattle husbandries expect the oral administration mainly to prevent respiratory diseases and/or liver abscesses [77,78]. Moreover, wide-spectrum antibiotics are administered intramammary during the non-lactating period, to prevent/reduce the incidence of mastitis in lactating animals [72]. Antibiotics usage for prophylactic purposes has been practiced in the pig industries. Pigs are generally reared in balanced groups according to age, size and weight, facilitating the massive antibiotic treatment, especially in the most susceptible periods [79,80]. Such periods span from birth until the first lactation; in this period animals are, indeed, subjected to stressful and potentially infective practices such as the cut of the umbilical cord, cut of the tail and trimming of the teeth. Moreover, the vaccination and castration periods are considered as susceptible ones; thus, the administration of wide-spectrum antibiotics helps reducing the overall risk of infection [72,81,82].

Further usage of antibiotics in veterinary medicine is related to the treatment of infective diseases. Ideally, therapeutic interventions are designed following accurate pathogen identification and its antimicrobial susceptibility test (e.g., antibiogram) [83,84]. However, it is a common practice to extend the antimicrobial treatment to the whole livestock flock, on the attempt to limit the pathogen spread, resulting in an overuse of the antibiotics since uninfected animals are also subjected to the antibiotic treatment [72]. Most recurrent infections in the cattle husbandries for meat production are related to shipping fever, bovine pneumonia and diarrhea, requiring massive usage of common antibiotics, such as penicillin, quinolones, gentamicin, and tylosin [85]. Furthermore, wide-spectrum antibiotics are commonly administered for the treatment of liver infections whilst narrow-spectrum antibiotics, such as beta-lactams, are the first choice for the treatment of mastitis of streptococcal origin [86]. Globally, the therapeutic usage of antibiotics in the pig industry is rather low as compared with the antimicrobial usage as growth-promoting factor or prophylactic purposes [87]. Most common infections in the pig husbandries are related to the gastrointestinal tract and are associated with extensive usage of penicillin, tetracycline, aminoglycosides, and quinolones. Moreover, large quantities of tiamulin, lincomycin, and enrofloxacin are used for the treatment of enzootic pneumonia other than swine dysentery and ileitis [72,81,82]. In poultry, antibiotics for therapeutic purposes are generally provided with the drinking water. Penicillins, aminoglycosides, tetracyclines, macrolides, and a combination of sulfonamide/trimethoprim are usually employed [88,89]. Similar classes of antibiotics are also used in sheep and goat production husbandries [90].

## 6. Microorganisms Involved in Relevant Domestic Animal Infections

Acknowledged the fine line while distinguishing between veterinary- and human- pathogens, the following paragraphs aim at describing the major bacterial specimens responsible of relevant domestic animal infections and their contribution to the overall AMR onset and diffusion (Table 1).

### 6.1. Campylobacter spp.

*Campylobacter spp* are a group of spiral-shaped Gram-negative bacteria responsible for gastrointestinal infections in several domestic animals, such as cattle, chicken, turkey, pig, sheep, and pets, including dogs and cats [91,92]. The prevalent aetiologic agent is *Campylobacter jejuni*, but all Campylobacter isolates have demonstrated resistance against one or more antimicrobial compound(s) including quinolones, macrolides, lincosamides, chloramphenicol, aminoglycosides, tetracycline, β-lactams, cotrimoxazole, and tylosin [93,94,95]. The resistance patterns observed in these specimens are diverse, and change depending on the geographic area and the source of isolation. Accentuated increase in fluoroquinolone resistance pattern has been registered across the five continents with some Northern Europe countries scoring a further increased isolation rates of fluoroquinolone-resistant *Campylobacter spp*. of poultry origin; most likely due to the increased usage of the fluoroquinolones enrofloxacin in the poultry industry [96,97]. Fluoroquinolones resistance in *Campylobacter spp*. is mostly due to point mutation on the quinolone resistance-determining region (QRDR) of DNA-gyrase A. Further resistance is provided by the accumulation of point mutation in the DNA-gyrase A encoding region, as well as by the multidrug efflux pump, CmeABC, that synergistically cooperate to enhance resistance by reducing the load of the antimicrobial compound within the bacterial cell [98,99,100,101]. Besides fluoroquinolones, increased resistance against macrolides has also been observed recently with surveillance data indicating *Campylobacter coli* isolates as being more resistant to macrolides than *C. jejuni* ones [96]. Macrolides resistance is mainly due to enzyme-mediated methylation or point mutation in the ribosomal target [102,103]. In addition, as for the fluoroquinolone resistance, good efflux capabilities are associated with a significantly enhanced resistance, suggesting a synergistic activity between the efflux pumps and the modification of the macrolide’s targets. [104,105,106,107].

### 6.2. Salmonella spp.

Salmonellosis is the infection provoked by the bacteria belonging to the genus Salmonella. These are rod-shaped Gram-negative bacteria commonly found in the gastrointestinal tract of the humans and other animals [108]. Pathogenic strains might infect a broad range of animals, including humans, poultry, and swine but cattle, horse, cats, and dogs are also frequently infected [109,110]. The association between antibiotic usage and the insurgence of antibiotic-resistant Salmonella strains at farm level as well as its consequences for human health is being extensively investigated since a relatively long time [111,112,113]. Monitoring data identified Salmonella as resistant to various antimicrobial agents, such as tetracyclines, sulfonamides, streptomycin, kanamycin, chloramphenicol, and some of the β-lactams [114,115,116]. Besides the “traditional” antibiotic classes, it is worthy of note that an increasing resistance trend is being registered against “novel” classes or the combination of antibiotics classes, such as amoxicillin/clavulanic acid, nalidixic acid, and ceftriaxone.

Drug resistance mechanisms activated by *Salmonella spp*. relies on the accumulation of mutations at the level of the genome encoding quinolone targets. Unlike *Campylobacter spp.* a plurality of point mutation is required to achieve a significant “resistance” status but multiple target regions are contributing to AMR: DNA Gyrase (GyrA and GyrB genes) and topoisomerase IV (ParC and ParE genes) are the most influent target for obtaining fluoroquinolones resistant strains [8,117,118,119]. Other mutations leading to the overexpression of multidrug efflux pumps AcrAB-TolC also have a pivotal role in the multidrug resistance of *Salmonella spp.* [120]. Moreover, changes in the outer membrane proteins also have a role in the AMR mechanisms [121]. In addition to the so-called classical quinolone resistance phenotype, it has been demonstrated, in nontyphoidal *Salmonella enterica* strains, the presence of highly mobile quinolone resistance (qnr) genotype, where a plasmid containing *qnr* genes confers the nontypical quinolone resistance phenotype in *S. enterica* isolates. This results in bacterial survival even at elevated quinolone concentrations, and therefore, strains carrying *qnr* alleles may be able to expand and propagate AMR during antibiotic treatment [122]. Moreover, systemic presence of Salmonella pathogenicity island-2 (SPI-2) or SPI-1 in gut-associated tissues enable the survival of Salmonella persisters that, in turn, have shown to favor the long-term plasmid dissemination among recipient species of gut microbiota, resulting in an enormous rearrangement and propagation of the AMR genetic elements [123].

### 6.3. Staphylococcus spp.

Staphylococcus spp. are coccal-shaped Gram-positive bacteria that colonize the skin, nares, and the mucosal membranes [124]. Although known as commensals, some bacterial strains are responsible for important infectious diseases in humans and a broad plethora of domestic animals [125]. Penicillin-resistant strains of *S. aureus* have been isolated already back in the early 1950s, suggesting a role of the extensive usage of antibiotics for zootechnical purposes in the production and dissemination of AMR traits. The initial hypothesis suggests a possible human origin of the strain and imputed to the host switching the acquisition of methicillin resistance trait, indicating the porcine feed supplemented with antibiotics as the most plausible AMR cause. Subsequent studies rejected this hypothesis by proving an independent emergence of the human pathogenic and the animal pathogenic strains [72,126]. To date, methicillin-resistant are the strains predominantly isolated from animals and animal by-products. Resistance mechanisms are conferred by the acquisition of the staphylococcal cassette chromosome *mec* (SCC*mec*), containing diverse *mec* genes responsible, among the other functions, of antimicrobial resistance. Being enclosed in mobile genetic elements, AMR genes are likely to be efficiently transmitted to recipient commensal, harboring the same ecological niches via horizontal gene transfer [127,128,129,130,131].

### 6.4. Enterococcus spp.

Enterococcus spp. are a heterogeneous group of bacteria commonly found as a member of the gut microbiota of humans and farm animals [108,132,133]. Enterococcus strains are capable of surviving to the harsh environmental condition such as extreme temperatures, various pH conditions, and halophilic environment. Moreover, *Enterococcus spp.* isolated from animals and food products have shown resistance to various antibiotics. In this view, Enterococcus strains are intrinsically resistant to many of the commonly used antimicrobial classes such as vancomycin, aminoglycosides and penicillin. Here, resistance is mediated by the product of the *lsa* gene, even though the molecular details are still poorly understood. Enterococcus spp. are capable of rapid acquisition of AMR traits when treated with antimicrobial compounds [134]. Introduction of chloramphenicol, erythromycin and tetracyclines was rapidly followed by the insurgence of resistant Enterococcus strains at a prevalence as high to pose the need to revoke the use of some antibiotic compounds in the clinical practice [135]. Nevertheless, *Enterococcus spp.* of animal and food origin is not intended as a direct risk source for humans. They are rather considered indirect risk by means of AMR transmission to taxonomically-related humans-adapted specimens, as already observed with the transmission of vancomycin resistance in *S. aureus* and tetracycline and erythromycin resistance in *Listeria monocytogenes* [72,136,137].

Besides the bacterial specimens reported above, many other bacteria linked to the veterinary medicine context are indicated as responsible of AMR dissemination; these include the heterogeneous ensemble of Gram-negative bacteria producing extended-spectrum beta-lactamases (ESBL), a plasmid-encoded feature that confers resistance against β-lactams such as third- or fourth-generation cephalosporins and monobactams. The incidence of ESBL-producing *Escherichia coli* in domestic animals such as cattle, broiler chicken, and pig is steadily increasing, suggesting a further potential risk of AMR trait dissemination by means of the animals-food-environment circle [72,138].

## 7. Latest Frontiers on Veterinary AMR Research

The initial investigations on the bacterial resistome were restricted to the sole genes responsible of conferring both direct and indirect tolerance against antimicrobial compounds, with a particular focus on those employed in the clinical practice [139,140]. It is nowadays well established that a pan-microbial approach targeting multiple levels of the biological macromolecules (i.e., DNA, RNA, proteins, and metabolites) enables access to a comprehensive investigation of the onset, development and transfer of the resistance mechanisms [141]. Moreover, a thorough understanding of interspecies communication is of paramount importance for elucidating how environmental factors trigger the resistance mechanisms and their molecular details. In this light, a wide variety of ecological niches harboring complex microbial community are the object of resistome-aimed studies, including the resistome investigation in manure, soil, wastewater, and animal-by food products from a variety of productive processes [3,142,143].

Methods employed for investigating AMR onset and diffusion evolved from the traditional culture-based approach to the latest omics technologies [144]. Based on a holistic approach, omics and meta-omics sciences represent attractive and powerful tools to investigate the functions and dynamics of the microorganisms (and microorganisms consortia) harbored in the different ecological niches [145].

Genomics is the discipline that studies the microbial genome, elucidating the potential metabolic functions by providing a catalogue of all the genes and genetic elements (e.g., plasmids and other genetic elements), besides enabling the deep taxonomic classification of the microorganisms and/or the consortia of microorganisms under study. Moreover, genomics has a pivotal influence on the other omics disciplines by constantly increasing the sequences database repository used as a reference [146,147,148,149]. A whole-genome sequencing approach has been recently employed in a survey on poultry-associated Salmonella. The study indicates that 60% of the poultry isolates were multidrug-resistant, as supported by the identification of chromosomal Single Nucleotide Polimorphysms (SNPs) and the identification of diverse mobile genetic elements. Furthermore, sequence analysis reports a novel streptomycin-azithromycin resistance island and an uncharacterized version of the SGI1, confirming the poultry industry as a reservoir of AMR traits that, might be efficiently transmitted to other bacterial strains [150]. In a similar study, genomics investigation of the chicken livers indicated a high predominance (>88% of isolates tested) of multidrug-resistant E. coli. Alarmingly, all investigated samples showed lincomycin resistance, whereas resistance to other antimicrobial compounds was observed at a variable extent [151]. In the context of microbial consortia, metagenomics depicts the overall genes and mobile genetic elements array enabling a clear evaluation of the genetically encoded AMR traits. A recent metagenomic investigation performed on the bacterial species harboring the rumen microbiota highlights a high prevalence of tetracycline resistance genes and their inclusion into novel integrative and conjugative elements, underlying the importance of the microbiota for the dissemination of the AMR and how bacterial evolutionary machinery of diverse specimens cooperate to guarantee bacterial survival [152]. Next-generation sequencing technologies are also being extensively applied to edible animal-by products enabling the monitoring of AMR diffusion and/or the identification of novel AMR traits. Genomic investigation of bovine milk samples enabled thorough analysis of 10 different *Arcobacter butzleri* strains, highlighting that 100% of isolates tested were resistant to fluoroquinolones and tetracycline; 90% of strains to rifampicin and cephalosporins and a variable prevalence of resistance to other antibiotics. Moreover, the study revealed that 50% of strains display four mutations in the *Mycobacterium tuberculosis* katG gene conferring resistance to isoniazid, providing evidence of the role of the animal-by product in the transmission of AMR [153].

Using the same next-generation sequencing technologies, transcriptomics complements the genomics information by highlighting the genome expression profile following specific conditions at the moment of sampling. Comparative analysis of the expression profiles of selected microorganisms and/or microbiota are commonly performed in the context of AMR studies to underline changes triggered by the antibiotic administration [154,155,156]. A recent metatranscriptomics assessment of the diversity and expression levels of resistance genes in the gut microbiome of birds with aquatic habits revealed a strong effect of the environment while eliciting the expression of AMR traits [157]. Moreover, transcriptomics approach was employed in the comparative evaluation of the expression profile of methicillin-resistant *Staphylococcus aureus* isolated from pork [158]. The study highlights that administration of subinhibitory doses of antibacterial compounds provoke alteration of the transcriptional profile, with genes related to membrane transport, amino acids, carbohydrate, and energy metabolism being differentially expressed.

Proteomics (and metaproteomics) are the disciplines targeting the ensemble of proteins expressed by the microorganisms (or microbiota), at the moment and condition of sampling [159,160,161]. Along with metabolomics, proteomics provides the most realistic picture of the key effectors that directly mediate the metabolic functions operated by the organisms [161], resulting in a very powerful tool to monitor the metabolism of antibiotic compounds and how the antimicrobial compounds trigger the microbial adaptive machinery responsible of the onset and diffusion of the AMR. Proteomics was employed to comparatively evaluate the membrane and cytosolic proteome of multidrug-resistant E. coli isolated from a water buffalo [162]. The study indicates differentially expressed proteins under multidrug resistance conditions. Functional classification of the differentially expressed proteins indicates intercellular communication mechanisms such as quorum sensing as being involved in the multidrug resistance, opening new avenues for future research projects. Following a similar approach, comparative proteomics investigation was employed for studying the outer membrane proteome of the kanamycin-resistant *E. coli* K-12 strain. The study depicts a higher expression of TolC, Tsx and OstA proteins, whereas MipA, OmpA, FadL, and OmpW were down-regulated in kanamycin-resistant strain [163]. Another study made use of proteomics to elucidate the mechanisms of enrofloxacin resistance in canine *E. coli* isolates [164]. MS-based analysis of the differentially expressed proteins shows an increased involvement of the resistant strain in DNA protection and oxidative stress response, indicating an active effort of the bacteria in counterbalancing the antibiotic effects. On this basis, one might conclude that each antibiotic compound elicits a specific response which, in turn, is mirrored in a characteristic proteome identification. However, a common “proteome signature” indicative of the ongoing antimicrobial resistance activity has been hypothesized since proteins involved in the energy and nitrogen metabolism, protein and nucleic acid synthesis, glucan biosynthesis, and stress response are generally affected [165,166].

Metabolomics provides a snapshot of the metabolites array produced by a given microorganism and/or microbial community; enabling a comprehensive investigation of the effective metabolism and the comparative evaluation following antimicrobial treatment [167,168,169]. Although of extreme importance, the bioinformatic data analysis of the metabolomic investigation is rather hard, discouraging a massive usage of such discipline, or limiting its employment as a tool to confirm previous outcomes of third disciplines. Nevertheless, a recent study of Lin et al. employed a metabolomic approach to compare the metabolome of two susceptible and two multidrug-resistant E. coli strain. Interestingly the metabolic profile identified enabled a clear distinction between the resistant and susceptible strains. The functional analysis described the resistant strains as more concerned in the biosynthesis of amino acids, biosynthesis of phenylpropanoids and purine metabolism [170]. Altogether, integration of the metabolomics data along with the proteomics outcomes might represent a novel approach for the investigation and prediction of the AMR traits [171].

## 8. Future Perspective for Preventing Insurgence and/or Development of AMR

The global depiction of the antibiotic resistance is becoming alarming and the severity of this emergence is destined to worsen in the near future. Antibiotic resistance of animal origin has been proven to contribute to AMR in a significant manner and, in concert with human medicine and environment, actions need to be taken to face this overwhelming issue. In the first instance, a judicious usage and management of the antibiotics is certainly helpful in delaying the phenomenon of antibiotics resistance. Although restrictive measures have been employed by the European Union (EU) in the veterinary field such as the antibiotic ban for growth-promoting purposes and the traceability of the antibiotic prescription, no significant improvement have been registered underlining, once again, the need for joint intervention between the “diverse” sectors and a worldwide effort in the acquisition and/or enforcement of regulatory measures.

Besides the cautious use of antibiotics, the adoption of adequate prophylactic interventions such as immunization programs is warmly required. Although incapable of cross-protection against specific pathogen(s) as prophylactic usage of antibiotic does, immunization guarantee a reduced pool of the infective agents and reduce the AMR transmission. Nevertheless, adoption of efficient vaccination programs cannot be considered as a fully resolutive measure, and complementary alternatives are needed for the nontherapeutic management of animal husbandry. A successful alternative to prevent infectious diseases is represented by the adoption of suitable probiotics and/or prebiotics that act directly on the gut microbiota. Optimized gut microbiota composition and functionalities improve the whole immune system and its protective performances other than ensuring a better feed conversion rate and increased competence against pathogen colonization [108,132,133,172]. In this light, several research lines are ongoing to elucidate the dynamic composition and mechanisms of the gut microbiota and provide guidance in the choice of the most suitable probiotic(s) and/or prebiotics.

Another way for preventing infectious disease relies on the use of bioactive peptides at bacteriostatic or bactericidal effect. Nisin A, a naturally produced bacteriocin by lactic bacteria have shown promising results, since harmless for the mammalian species and rather efficient in the control of pathogenic specimens. Cumulative evidence proved the efficacy of bacteriocins also in the food industry. Indigenous microflora of raw milk control pathogenic bacteria through the production of diverse bacteriocins, including Nisin A [143]. Studies evaluating bacteriocins stability in vivo, as well as the most suitable delivery strategy, are now under investigation.

Predatory bacteria might also represent a promising alternative to antimicrobials. *Bdellovibrio bacteriovorus* and *Micavibrio aeruginovorus* demonstrated the effective reduction of pathogens such as *Acinetobacter baumannii*, *E. coli*, *Klebsiella pneumoniae*, *Pseudomonas aeruginosa*, and *Pseudomonas putida*. Nevertheless, bacteria predatory are not capable of discriminating between antibiotic-resistant and non-antibiotic resistant strain and might be a potential threat for the beneficial commensal flora; thus, hindering a direct adoption of this alternative.

Besides scientific research to improve knowledge and the discover novel classes of antimicrobials and alternatives, appropriate legislative measures and enforcement are required to guarantee fair “behavior” and ensure public health. At this purpose, restrictive regulations have been relatively recently imposed to veterinarians in the EU countries. Nevertheless, the issue does not seem to be ameliorated, indicating an intrinsic bias while considering individual responsibilities (e.g., veterinary medicine instead of human medicine), leaving space to the holistic vision that everyone who used antibiotics has co-share contribute in the warning phenomenon of antibiotic resistance.

## Figures and Tables

**Figure 1 ijms-21-01914-f001:**
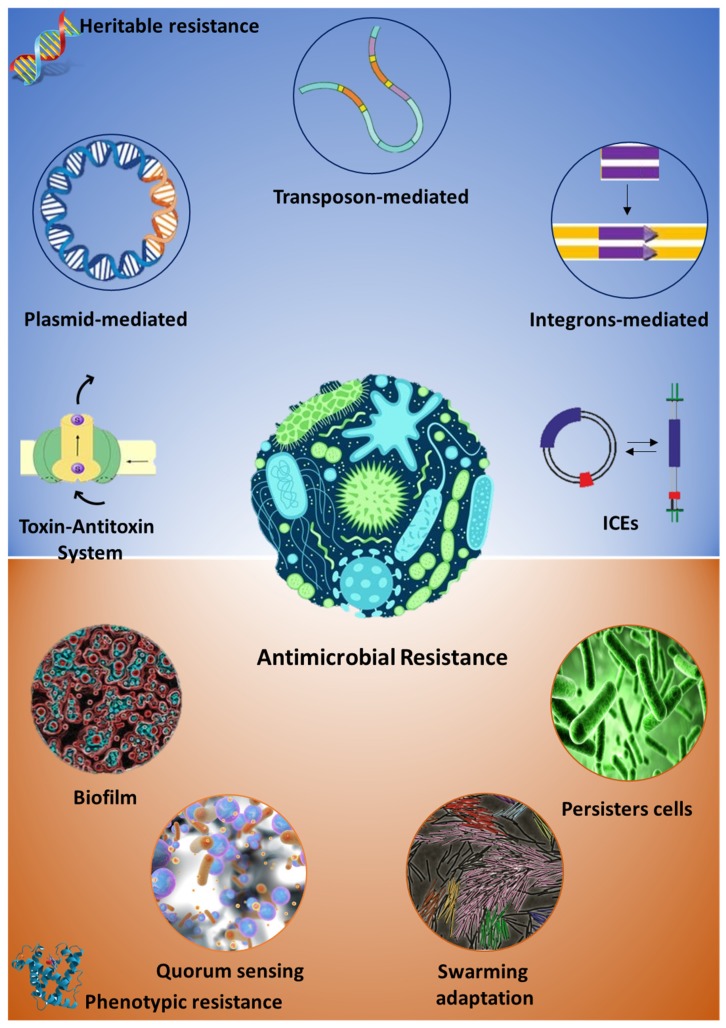
Molecular mechanisms driving antimicrobial resistance. Image depicts heritable (blue framed) and phenotypic (orange framed) mechanisms of antimicrobial resistance in bacteria. ICES = Integrative and Conjugative Elements.

**Table 1 ijms-21-01914-t001:** Major microorganisms responsible of relevant infective disease and AMR onset and diffusion.

Resistant Bacteria	Isolation Source	Antimicrobial Compound(s)	Major Resistance Mechanism
*Campylobacter spp.*	Cattle Chickens Turkey Pig Sheep Dog Cat Horse	Quinolones Macrolides Lincosamides Chloramphenicol Aminoglycosides Tetracycline β-lactams Cotrimoxazole Tylosin	Point mutation on *GyrA* gene *CmeABC* multidrug efflux pump Methylation of ribosomal target Point mutation in the ribosomal target
*Salmonella spp.*	Human Chicken Turkey Pig Cat Dog Horse	Tetracyclines Sulfonamides Streptomycin Kanamycin Chloramphenicol β-lactams Amoxicillin/clavulanic acid Nalidixic acid Ceftriaxone	Multiple point mutations on *GyrA* and *GyrB*, *parC*, and *parE* genes *AcrAB-TolC* multidrug efflux pumps Changing outer membrane proteins *qnr* genes-containing plasmid Salmonella Pathogenicity Island-1 and -2
*Staphylococcus spp.*	Human Farm animals Dog Cat Horse	Penicillin Methicillin Vancomycin	Staphylococcal Cassette Chromosome-mec genes (*SCC-mec*)
*Enterococcus spp*	Human Farm animals	Vancomycin, Aminoglycosides Penicillin Chloramphenicol Erythromycin Tetracyclines	*lsa* gene
Other Gram-negative	Human Farm animals Pet	β-lactams	Extended-spectrum β-lactamases

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
