# Peer review of "Antimicrobial Resistance in Veterinary Medicine: An Overview"

_ijms, 2020, doi:10.3390/ijms21061914_

Round 1

Reviewer 1 Report

The manuscript details a current and relevant topic as antimicrobial resistace is a global challange worldwide both in human medicine and in veterinary field.

Present manuscript is written based on current literature, and gives a detailed overview about this topic.

In my opinion this manuscript is suitable for publication.

Author Response

The manuscript details a current and relevant topic as antimicrobial resistace is a global challange worldwide both in human medicine and in veterinary field.

Present manuscript is written based on current literature, and gives a detailed overview about this topic.

In my opinion this manuscript is suitable for publication.

We thank the Reviewer#1 for the kind revision. We are pleased by reading that other experts appreciated our literature review and the resulting manuscript. We wish that, following the adjustments suggested by the other Reviewers, the manuscript will be suitable for publication in the International Journal of Molecular Sciences.

Reviewer 2 Report

Dear Authors, Your manuscript provides excellent overview of history and current state of antimicrobial resistance in veterinary drugs. It also highlights disparity in attention paid to the issue compared to human resistance. However, I found the writing needs review of grammar through out the manuscript.

Author Response

Dear Authors, your manuscript provides excellent overview of history and current state of antimicrobial resistance in veterinary drugs. It also highlights disparity in attention paid to the issue compared to human resistance. However, I found the writing needs review of grammar through out the manuscript.

We gratefully acknowledge Reviwer#2 for the revision of our manuscript and the constructive suggestion. The revised version of the manuscript has been thoroughly reviewed by a native English-speaking colleague and improved accordingly.

Reviewer 3 Report

The paper demonstrates the dire issue with antimicrobial resistance in veterinary medicine. The document makes reference to the use of current human antibiotics in the veterinary arena and how this is driving AMR. The review covers many of the methods bacteria have/utilise to acquire resistance and describe these in detail. This is a very good review however; it requires an extensive revision of the English language and grammar as in its current form it is difficult to read. The document highlights some of the areas but a comprehensive review is required.

Upon addressing this, I would have no hesitation in recommending this review for publication.

Author Response

The paper demonstrates the dire issue with antimicrobial resistance in veterinary medicine. The document makes reference to the use of current human antibiotics in the veterinary arena and how this is driving AMR. The review covers many of the methods bacteria have/utilise to acquire resistance and describe these in detail. This is a very good review however; it requires an extensive revision of the English language and grammar as in its current form it is difficult to read. The document highlights some of the areas but a comprehensive review is required.

Upon addressing this, I would have no hesitation in recommending this review for publication.

We gratefully acknowledge Reviewer #3 for the meticulous revision of our manuscript. We are pleased by reading that our manuscript aroused the interest of other experts in this field. The revised version of the manuscript has been extensively reviewed in the language grammar, sentence structure and text fluency, improving overall manuscript readability.

Round 2

Reviewer 3 Report

The authors have now rectified the English and grammar of the document sufficiently to allow for publication in its current form. I believe that this is a very useful document to highlight the crossover and many issues associated with the utility of antibiotics in different species but remains somewhat hidden from popular discussion. This will aid in the revealing and unmasking this issue so we can engage with it to address the dire need in AMR research.